# Learning Off-Policy with Online Planning

**Harshit Sikchi,**[*] **Wenxuan Zhou, David Held**
Robotics Institute
Carnegie Mellon University
`{hsikchi, wenxuanz, dheld}@cs.cmu.edu`

**Abstract:** Reinforcement learning (RL) in low-data and risk-sensitive domains requires performant and flexible deployment policies that can readily incorporate constraints during deployment. One such class of policies are the semi-parametric H-step lookahead policies, which select actions using trajectory optimization over a dynamics model for a fixed horizon with a terminal value function. In this work, we investigate a novel instantiation of H-step lookahead with a *learned* model and a terminal value function learned by a *model-free off-policy* algorithm, named Learning Off-Policy with Online Planning (LOOP). We provide a theoretical analysis of this method, suggesting a tradeoff between model errors and value function errors and empirically demonstrate this tradeoff to be beneficial in deep reinforcement learning. Furthermore, we identify the "Actor Divergence" issue in this framework and propose Actor Regularized Control (ARC), a modified trajectory optimization procedure. We evaluate our method on a set of robotic tasks for Offline and Online RL and demonstrate improved performance. We also show the flexibility of LOOP to incorporate safety constraints during deployment with a set of navigation environments. We demonstrate that LOOP is a desirable framework for robotics applications based on its strong performance in various important RL settings. Project video and details can be found at hari-sikchi.github.io/loop.

**Keywords:** Reinforcement Learning, Trajectory Optimization, Safety

## 1 Introduction

Off-policy reinforcement learning algorithms have been widely used in many robotic applications due to their sample efficiency and their ability to incorporate data from different sources [1, 2, 3, 4]. Model-free off-policy algorithms sample transitions from a replay buffer to learn a value function and then update the policy according to the value function [5, 6]. Thus, the performance of the policy is highly dependent on the estimation of the value function. However, learning an accurate value function from off-policy data is challenging especially in deep RL due to a variety of issues, such as overestimation bias [7, 8], delusional bias [9], rank

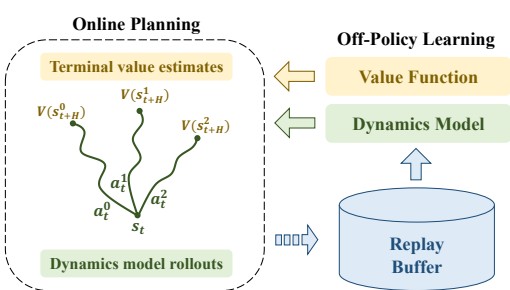

Figure 1: Overview of LOOP: A learned dynamics model is utilized for Online Planning with a terminal value function. The value function is learned via a model-free off-policy algorithm.

loss [10], instability [11], and divergence [12]. Another shortfall of model-free off-policy algorithms in continuous control is that the policy is usually parametrized by a feedforward neural network which lacks flexibility during deployment.

Previous works in model-based RL have explored different ways of using a dynamics model to improve off-policy algorithms [13, 14, 15, 16, 17]. One way of incorporating the dynamics model is to use H-step lookahead policies [18]. At each timestep, H-step lookahead policies rollout the dynamics model H-step into the future from the current state to find an action sequence with the highest return. Within this trajectory optimization process, a terminal value function is attached to the end of the rollouts to provide an estimation of the return beyond the fixed horizon. This way of online

---

[*]Currently at The University of Texas at Austin, Email: hsikchi@utexas.edu

5th Conference on Robot Learning (CoRL 2021), London, UK.

planning offers us a degree of explainability missing in fully parametric methods while also allowing us to take constraints into account during deployment. Previous work proves faster convergence with H-step lookahead policies in tabular setting [18] or showed improved sample complexity with a *ground-truth* dynamics model [19]. However, the benefit of H-step lookahead policies remains unclear under an *approximate model* and an *approximate value function*. Additionally, if H-step lookahead policies are used during the value function update [19], the required computation of value function update will be significantly increased.

In this work, we take this direction further by studying H-step lookahead both theoretically and empirically with three main contributions. **First**, we provide a theoretical analysis of H-step lookahead under an *approximate model* and *approximate value function*. Our analysis suggests a trade-off between model error and value function error, and we empirically show that this tradeoff can be used to improve policy performance in Deep RL. **Second**, we introduce Learning Off-Policy with Online Planning (LOOP) (Figure 1). To avoid the computational overhead of performing trajectory optimization while updating the value function as in previous work [19], the value function of LOOP is updated via a parameterized actor using a model-free off-policy algorithm ("Learning Off-Policy"). LOOP exploits the benefits of H-step lookahead policies when the agent is deployed in the environment during exploration and evaluation ("Online Planning"). This novel combination of model-based online planning and model-free off-policy learning provides sample-efficient and computationally-efficient learning. We also identify the "Actor Divergence" issue in this combination and propose a modified trajectory optimization method called Actor Regularized Control (ARC). ARC performs implicit divergence regularization with the parameterized actor through Iterative Importance Sampling.

**Third**, we explore the flexibility of H-step lookahead policies for improved performance in offline RL and safe RL, which are both important settings in robotics. LOOP can be applied on top of various offline RL algorithms to improve their evaluation performance. LOOP's semiparameteric behavior policy also allows it to easily incorporate safety constraints during deployment. We evaluate LOOP on a set of simulated robotic tasks including locomotion, manipulation, and controlling an RC car. We show that LOOP provides significant improvement in performance for online RL, offline RL, and safe RL, which makes it a strong choice of RL algorithm for robotic applications.

## 2  Related Work

**Model-based RL** Model-based reinforcement learning (MBRL) methods learn a dynamics model and use it to optimize the policy. State-of-the-art model-based RL methods usually have better sample efficiency compared to model-free methods while maintaining competitive asymptotic performance [20, 13]. One approach in MBRL is to use trajectory optimization with a learned dynamics model [17, 21, 22]. These methods can reach optimal performance when a large enough planning horizon is used. However, they are limited by not being able to reason about the rewards beyond the planning horizon. Increasing the planning horizon increases the number of trajectories that need to be sampled and incurs a heavy computational cost.

Various attempts have been made to combine model-free and model-based RL. GPS [23] combines trajectory optimization using analytical models with the on-policy policy gradient estimator. MBVE [15] and STEVE [16] use the model to improve target value estimates. Approaches such as MBPO [13] and MAAC [24] follow Dyna-style [25] learning where imagined short-horizon trajectories are used to provide additional transitions to the replay buffer leveraging model generalization. Piché et al. [26] use Sequential Monte Carlo (SMC) to capture multimodal policies. The SMC policy relies on combining multiple 1-step lookahead value functions to sample a trajectory proportional to the unnormalized probability $\exp(\sum_{i=1}^{H}(A(s,a)))$; this approach potentially compounds value function errors, in contrast to LOOP which uses single H-step lookahead planning for each state. POLO [19] shows advantages of trajectory optimization under *ground-truth* dynamics with a terminal value function. The value function updates involve additional trajectory optimization routines which is one of the issues we aim to address with LOOP. The computation of trajectory optimization in POLO is $\mathcal{O}(THN)$ while LOOP is $\mathcal{O}(TH)$ where $T$ is the number of environment timesteps, $H$ is the planning horizon, and $N$ is the number of samples needed for training the value function.

**Off-Policy RL** LOOP relies on a terminal value function for long horizon reasoning which can be learned effectively via model-free off-policy RL algorithms. Off-policy RL methods such as SAC [5]

and TD3 [6] use the replay buffer to learn a Q-function that evaluates a parameterized actor and then optimize the actor by maximizing the Q-function. Off-policy methods can be modified to be used for Offline RL problems where the goal is to learn a policy from a static dataset [27, 8, 28, 29, 30, 31, 32]. MBOP [33], a recent model-based offline RL method, leverages planning with a terminal value function, but the value function is a Monte Carlo evaluation of truncated replay buffer trajectories, whereas in LOOP the value function is trained for optimality under the dataset.

## 3 Preliminaries

A Markov Decision Process (MDP) is defined by the tuple $(\mathcal{S}, \mathcal{A}, p, r, \rho_0)$ with state-space $\mathcal{S}$, action-space $\mathcal{A}$, transition probability $p(s_{t+1}|s_t, a_t)$, reward function $r(s, a)$, and initial state distribution $\rho_0(s)$. In the infinite horizon discounted MDP, the goal of reinforcement learning algorithms is to maximize the return for policy $\pi$ given by $J^\pi = \mathbb{E}_{a_t \sim \pi(s_t), s_0 \sim \rho_0}[\sum_{t=0}^{\infty} \gamma^t r(s_t, a_t)]$.

**Value functions:** $V^\pi : \mathcal{S} \to \mathbb{R}$ represents a state-value function which estimates the return from the current state $s_t$ and following policy $\pi$, defined as $V^\pi(s) = \mathbb{E}_{a_t \sim \pi(s_t)}[\sum_{t=0}^{\infty} \gamma^t r(s_t, a_t)|s_0 = s]$. Similarly, $Q^\pi : \mathcal{S} \times \mathcal{A} \to \mathbb{R}$ represents a action-value function, usually referred as a Q-function, defined as $Q^\pi(s, a) = \mathbb{E}_{a_t \sim \pi(s_t)}[\sum_{t=0}^{\infty} \gamma^t r(s_t, a_t)|s_0 = s, a_0 = a]$. Value functions corresponding to the optimal policy $\pi^*$ are defined to be $V^*$ and $Q^*$. The value function can be updated according to the Bellman operator $\mathcal{T}$:

$$\mathcal{T}Q(s_t, a_t) = r(s_t, a_t) + \mathbb{E}_{s_{t+1} \sim p, a_{t+1} \sim \pi_Q}[\gamma(Q(s_{t+1}, a_{t+1})] \tag{1}$$

where $\pi_Q$ is updated to be greedy with respect to $Q$, the current Q-function.

**Constrained MDP for safety:** A constrained MDP (CMDP) is defined by the tuple $(\mathcal{S}, \mathcal{A}, p, r, c, \rho_0)$ with an additional cost function $c(s, a)$. We define the cumulative cost of a policy to be $D^\pi = \mathbb{E}_{a_t \sim \pi(s_t), s_0 \sim \rho_0}[\sum_{t=0}^{\infty} \gamma^t c(s_t, a_t)]$. A common objective for safe reinforcement learning is to find a policy $\pi = \operatorname{argmax}_\pi J^\pi$ subject to $D^\pi \leq d_0$ where $d_0$ is a safety threshold [34].

## 4 H-step Lookahead with Learned Model and Value Function

Model-based algorithms often learn an approximate dynamics model $\hat{M}(s_{t+1}|s_t, a_t)$ using the data collected from the environment. One way of using the model is to find an action sequence that maximizes the cumulative reward with the learned model using trajectory optimization [35, 36, 37]. An important limitation of this approach is that the computation grows exponentially with the planning horizon. Thus, methods like [35, 17, 21, 38, 39] plan over a fixed, short horizon and are unable to reason about long-term reward. Let $\pi_H$ be such a fixed horizon policy:

$$\pi_H(s_0) = \operatorname*{argmax}_{a_0} \max_{a_1,..,a_{H-1}} \mathbb{E}_{\hat{M}}[R_H(s_0, \tau)] \text{, where } R_H(s_0, \tau) = \sum_{t=0}^{H-1} \gamma^t r(s_t, a_t) \tag{2}$$

where $\tau$ denotes the action sequence $a_{[0..H-1]}$. One way to enable efficient long-horizon reasoning is to augment the planning trajectory with a terminal value function. Given a value-function $\hat{V}$, we define a policy $\pi_{H,\hat{V}}$ obtained by maximizing the H-step lookahead objective:

$$\pi_{H,\hat{V}}(s_0) = \operatorname*{argmax}_{a_0} \max_{a_1,..,a_{H-1}} \mathbb{E}_{\hat{M}}\left[R_{H,\hat{V}}(s_0, \tau)\right] \tag{3}$$

$$\text{where } R_{H,\hat{V}}(s_0, \tau) = \sum_{t=0}^{H-1} \gamma^t r(s_t, a_t) + \gamma^H \hat{V}(s_H)$$

The quality of both the model $\hat{M}$ and the value-function $\hat{V}$ affects the performance of the overall policy. To show the benefits of this combination of model-based trajectory optimization and the value-function, we now analyze and bound the performance of the H-step look-ahead policy $\pi_{H,\hat{V}}$ compared to its fixed-horizon counterpart without the value-function $\pi_H$ (Eqn. 2), as well as the greedy policy obtained from the value-function $\pi_{\hat{V}} = \operatorname{argmax}_a \mathbb{E}_{s' \sim M(.|s,a)}\left[r(s, a) + \gamma \hat{V}(s')\right]$. Following previous work, we will construct the proofs with the state-value function $V$, but the proofs for the action-value function $Q$ can be derived similarly.

**Lemma 1.** *(Singh and Yee [40]) Suppose we have an approximate value function $\hat{V}$ such that $\max_s |V^*(s) - \hat{V}(s)| \leq \epsilon_v$. Then the performance of the 1-step greedy policy $\pi_{\hat{V}}$ can be bounded as:*

$$J^{\pi^*} - J^{\pi_{\hat{V}}} \leq \frac{\gamma}{1-\gamma}[2\epsilon_v] \tag{4}$$

**Theorem 1.** *(H-step lookahead policy) Suppose $\hat{M}$ is an approximate dynamics model with Total Variation distance bounded by $\epsilon_m$. Let $\hat{V}$ be an approximate value function such that $\max_s |V^*(s) - \hat{V}(s)| \leq \epsilon_v$. Let the reward function $r(s,a)$ be bounded by $[0,R_{max}]$ and $\hat{V}$ be bounded by $[0,V_{max}]$. Let $\epsilon_p$ be the suboptimality incurred in H-step lookahead optimization (Eqn. 3). Then the performance of the H-step lookahead policy $\pi_{H,\hat{V}}$ can be bounded as:*

$$J^{\pi^*} - J^{\pi_{H,\hat{V}}} \leq \frac{2}{1-\gamma^H}[C(\epsilon_m, H, \gamma) + \frac{\epsilon_p}{2} + \gamma^H \epsilon_v] \tag{5}$$

*where*

$$C(\epsilon_m, H, \gamma) = R_{\max} \sum_{t=0}^{H-1} \gamma^t t \epsilon_m + \gamma^H H \epsilon_m V_{max}$$

*Proof.* Due to the page limit, we defer the proof to Appendix A.1. We also provide extension of Theorem 1 under assumptions on model generalization and concentrability in Corollary 1 and Theorem 2 respectively in Appendix A. □

**H-step Lookahead Policy vs H-step Fixed Horizon Policy:** The fixed-horizon policy $\pi_H$ can be considered as a special case of $\pi_{H,\hat{V}}$ with $\hat{V}(s) = 0 \; \forall s \in \mathcal{S}$. Following Theorem 1, $\epsilon_{\hat{V}} = \max_s |V^*(s)|$ implies a potentially large optimality gap. This suggests that learning a value function that better approximates $V^*$ than $\hat{V}(s) = 0$ will give us a smaller optimality gap in the worst case.

**H-step lookahead policy vs 1-step greedy policy:** By comparing Lemma 1 and Theorem 1, we observe that the performance of the H-step lookahead policy $\pi_{H,\hat{V}}$ reduces the dependency on the value function error $\epsilon_v$ at least by a factor of $\gamma^{H-1}$ while introducing an additional dependency on the model error $\epsilon_m$. This implies that the H-step lookahead is beneficial when the value-function bias dominates the bias in the learned model. In the low data regime, the value function bias can result from compounded sampling errors [41] and is likely to dominate the model bias, as evidenced by the success of model-based RL methods in the low-data regime [33, 42, 13]; we observe this hypothesis to be consistent with our experiments where H-step lookahead offers large gains in sample efficiency. Further, errors in value learning with function approximation can stem from a number of reasons explored in previous work, some of them being Overestimation, Rank Loss, Divergence, Delusional bias, and Instability [7, 11, 6, 43, 10]. Although this result may be intuitive to many practitioners, it has not been shown theoretically; further, we demonstrate that we can use this insight to improve the performance of state-of-the-art methods for online RL, offline RL, and safe RL.

## 5 Learning Off-Policy with Online Planning

We propose Learning Off-Policy with Online Planning (LOOP) as a framework of using H-step lookahead policies that combines online trajectory optimization with model-free off-policy RL (Figure 1). We use the replay buffer to learn a dynamics model and a value function using an off-policy algorithm. The H-step lookahead policy (Eqn. 3) generates rollouts using the dynamics model with a terminal value function and selects the best action for execution. The underlying off-policy algorithm is boosted by the H-step lookahead which improves the performance of the policy during both exploration and evaluation. From another perspective, the underlying model-based trajectory optimization is improved using a terminal value function for reasoning about future returns. In this section, we discuss the Actor Divergence issue in the LOOP framework and introduce additional applications and instantiations of LOOP for offline RL and safe RL.

### 5.1 Reducing actor-divergence with Actor Regularized Control (ARC)

As discussed above, LOOP utilizes model-free off-policy algorithms to learn a value function in a more computationally efficient manner. It relies on actor-critic methods which use a parametrized actor $\pi_\theta$ to facilitate the Bellman backup. However, we observe that combining trajectory optimization and policy learning naively will lead to an issue that we refer to as "actor divergence": a different policy is used for data collection (H-step lookahead policy $\pi_{H,\hat{V}}$) than the policy that is used to learn

the value-function (the parametrized actor $\pi_\theta$). This leads to a potential distribution shift between the state-action visitation distribution between the parametrized actor $\pi_\theta$ and the actual behavior policy $\pi_{H,\hat{V}}$ which can lead to accumulated bootstrapping errors with the Bellman update and destabilize value learning [43]. One possible solution in this case is to use Offline RL [30]; however, in practice, we observe that offline RL in this setup leads to learning instabilities. We defer discussion on this alternative to the Appendix D.7. Instead, we propose to resolve the actor-divergence issue via a modified trajectory optimization method called Actor Regularized Control (ARC).

In ARC, we aim to constrain the action selection of the trajectory optimization to be close to the parametrized actor. We frame the following general constrained optimization problem for policy improvement [44]:

$$p_{opt}^\tau = \underset{p^\tau}{\arg\max}\, \mathbb{E}_{p^\tau}[L_{H,\hat{V}}(s_t,\tau)]\,,\ \text{s.t}\ D_{KL}(p^\tau||p_{prior}^\tau) \le \epsilon \tag{6}$$

where $L_{H,\hat{V}}(s_t,\tau)$ is the expected lookahead objective (Eqn. 3) under the learned model given by $L_{H,\hat{V}}(s_t,\tau) = \mathbb{E}_{\hat{M}}\left[R_{H,\hat{V}}(s_t,\tau)\right]$, starting from state $s_t$, $p^\tau$ is a distribution over action sequences $\tau$ of horizon H starting from $s_t$, and $p_{prior}^\tau$ is a prior distribution over such action sequences. We will use the parametrized actor to derive this prior in ARC. This optimization admits a closed form solution by enforcing the KKT conditions where the optimal policy is given by $p_{opt}^\tau \propto p_{prior}^\tau e^{\frac{1}{\eta}L_{H,\hat{V}}(s_t,\tau)}$ [45, 46, 47, 48], where $\eta$ is the lagrangian dual variable. The above formulation generalizes a number of prior work [5, 35, 45] (more details in Appendix B.3).

Approximating the optimal policy $p_{opt}^\tau$ as a multivariate gaussian with diagonal covariance $\hat{p}_{opt}^\tau = \mathcal{N}(\mu_{opt},\sigma_{opt})$, the parameters can be estimated using importance sampling under the proposal distribution $p_{prior}^\tau$ as:

$$\hat{p}_{opt}^\tau = \mathcal{N}(\mu_{opt},\sigma_{opt})\,,\ \mu_{opt} = \mathbb{E}_{\tau',\hat{M}}\left[\frac{p_{opt}^\tau(\tau')}{p_{prior}^\tau(\tau')}\tau'\right]\,,\ \sigma_{opt} = \mathbb{E}_{\tau',\hat{M}}\left[\frac{p_{opt}^\tau(\tau')}{p_{prior}^\tau(\tau')}(\tau'-\mu)^2\right] \tag{7}$$

where $\tau' \sim p_{prior}^\tau$. We use iterative importance sampling to estimate $\hat{p}_{opt}^\tau$ which is parameterized as a Gaussian whose mean and variance at iteration $m+1$ are given by the empirical estimate:

$$\mu^{m+1} = \frac{\sum_{i=1}^N[e^{\frac{1}{\eta}L_{H,\hat{V}}(s_t,\tau')}\tau']}{\sum_{i=1}^N e^{\frac{1}{\eta}L_{H,\hat{V}}(s_t,\tau')}}\,,\ \sigma^{m+1} = \frac{\sum_{i=1}^N[e^{\frac{1}{\eta}L_{H,\hat{V}}(s_t,\tau')}(\tau'-\mu^{m+1})^2]}{\sum_{i=1}^N e^{\frac{1}{\eta}L_{H,\hat{V}}(s_t,\tau')}} \tag{8}$$

where $\tau' \sim \mathcal{N}(\mu^m,\sigma^m)$ and $\mathcal{N}(\mu^0,\sigma^0)$ is set to $p_{prior}^\tau$. As long as we perform a finite number of iterations, the final trajectory distribution is constrained in total variation to be close to the prior as a result of finite trust region updates as shown in Lemma 2 in Appendix A.4.

To reduce actor divergence in LOOP, we constrain the action-distribution of the trajectory optimization to be close to that of the parametrized actor $\pi_\theta$. To do so, we set $p_{prior}^\tau = \beta\pi_\theta + (1-\beta)\mathcal{N}(\mu_{t-1},\sigma)$. The trajectory prior is a mixture of the parametrized actor and the action sequence from the previous environment timestep with additional Gaussian noise $\mathcal{N}(0,\sigma)$. Using 1-timestep shifted solution from the previous timestep allows to amortize trajectory optimization over time [33]. For online RL, we can vary $\sigma$ to vary the amount of exploration during training. For offline RL, we set $\beta = 1$ to constrain actions to be close to those in the dataset (from which $\pi_\theta$ is learned) to be more conservative.

## 5.2 Additional instantiations of LOOP: Offline-LOOP and Safe-LOOP

LOOP not only improves the performance of previous model-based and model-free RL algorithms but also shows versatility in different settings such as the offline RL setting and the safe RL setting. These potentials of H-step lookahead policies have not been explored in previous work.

**LOOP for Offline RL:** In offline reinforcement learning, the policy is learned from a static dataset without further data collection. We can use LOOP on top of an existing off-policy algorithm as a plug-in component to improve its test time performance by using the model-based rollouts as suggested by Theorem 1. Note that this is different from the online setting in the previous section in which LOOP also influences exploration. In offline-LOOP, to account for the uncertainty in the model and the Q-function, ARC optimizes for the following uncertainty-pessimistic objective similar to [49, 50]:

$$\text{mean}_{[K]}[R_{H,\hat{V}}(s_t,\tau)] - \beta_{pess}\text{std}_{[K]}[R_{H,\hat{V}}(s_t,\tau)] \tag{9}$$

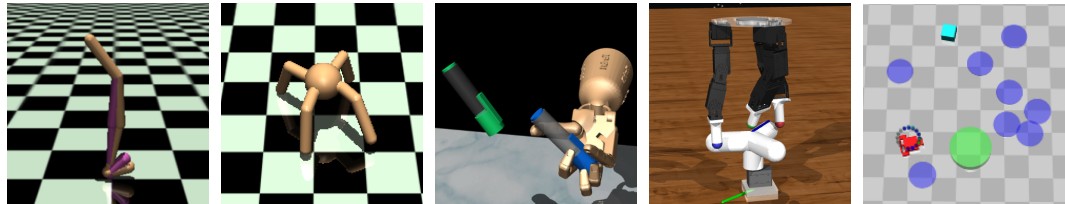

Figure 2: We evaluate LOOP over a variety of environments ranging from locomotion, manipulation to navigation including Walker2d-v2, Ant-v2, PenGoal-v1, Claw-v1, CarGoal1, etc.

where $[K]$ are the model ensembles, $\beta_{pess}$ is the pessimism parameter and $R_{H,\hat{V}}$ is the H-horizon lookahead objective defined in Eqn. 3.

**Safe Reinforcement Learning:** Another benefit of LOOP with its semi-parameteric policy is that we can easily incorporate (possibly non-stationary) constraints with the model-based rollout, while being an order of magnitude more sample efficient than existing safe model-free RL algorithms. To account for safety in the planning horizon, ARC optimizes for the following cost-pessimistic objective:

$$\text{argmax}_{a_t} \mathbb{E}_{\hat{M}}\left[R_{H,\hat{V}}(s_t, \tau)\right] \text{s.t.} \max_{[K]} \sum_{t=t}^{t+H-1} \gamma^t c(s_t, a_t) \leq d_0 \tag{10}$$

where $[K]$ are the model ensembles, $c$ is the constraint cost function and $R_{H,\hat{V}}$ is the H-horizon lookahead objective defined in Eqn. 3 and $d_0$ is the constraint threshold. For each action rollout, the worst-case cost is considered w.r.t model uncertainty to be more conservative. The pseudocode for modified ARC to solve the above constrained optimization is given in Appendix B.3.1.

## 6 Experimental Results

In the experiments, we evaluate the performance of LOOP combined with different off-policy algorithms in the settings of online RL, offline RL and safe RL over a variety of environments (Figure 2). Implementation details of LOOP and the baselines can be found in Appendix C.

### 6.1 LOOP for Online RL

In this section, we evaluate the performance of LOOP for online RL on three OpenAI Gym MuJoCo [51] locomotion control tasks: `HalfCheetah-v2`, `Walker-v2`, `Ant-v2` and two manipulation tasks: `PenGoal-v1`, `Claw-v1`. In these experiments, we use Soft Actor-Critic (SAC) [5] as the underlying off-policy method with the ARC optimizer described in Section 5.1. Further experiments on `InvertedPendulum-v2`, `Swimmer`, `Hopper-v2` and `Humanoid-v2` and more details on the baselines can be found in Appendix D.1 and Appendix C.2 respectively.

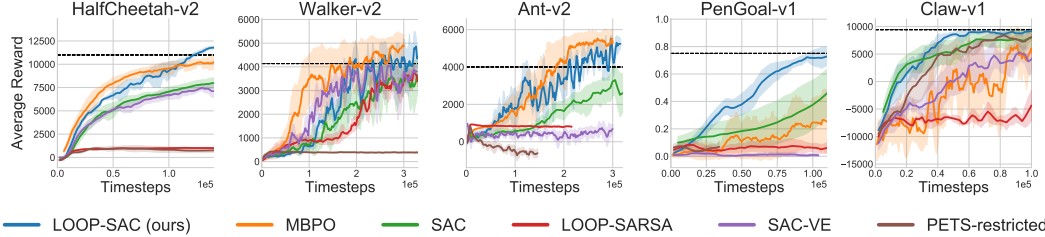

Figure 3: Comparisons of LOOP and the baselines for online RL. LOOP-SAC is significantly more sample efficient than SAC. It is competitive to MBPO for locomotion tasks and outperforms MBPO for manipulation tasks (PenGoal-v1 and Claw-v1). The dashed line indicates the performance of SAC at 1 million timesteps. Additional results on more environments can be found in Appendix D.1.

**Baselines:** We compare the LOOP framework against the following baselines: PETS-restricted, a variant of PETS [17] that uses trajectory optimization (CEM) for the same horizon as LOOP but without a terminal value function. LOOP-SARSA uses a terminal value function which is an evaluation of the replay buffer policy, similar to MBOP [33] in spirit. To compare with other ways of combining model-based and model-free RL, we also compare against MBPO [13] and SAC-VE. MBPO leverages the learned model to generate additional transitions for value function learning.

SAC-VE utilizes the model for value expansion, similar to MBVE [15] but uses SAC as the model-free component for a fair comparison with LOOP as done in [13]. We do not include comparison to STEVE [16] or SLBO [52] as they were shown to be outperformed by MBPO, and perform poorly compared to SAC in Hopper and Walker environments [13]. We were unable to reproduce the results for SMC [26] due to missing implementation. We did not include POLO here due several reasons. An extended discussion can be found in Appendix D.2.

**Performance:** From Figure 3, we observe that LOOP-SAC is significantly more sample efficient than SAC, the underlying model-free method used to learn a terminal value function. LOOP-SAC also scales well to high-dimensional environments like `Ant-v2` and `PenGoal-v1`. PETS-restricted performs poorly due to myopic reasoning over a limited horizon $H$. SAC-VE and MBPO represent different ways of incorporating a model to improve off-policy learning. LOOP-SAC outperforms SAC-VE and performs competitively to MBPO, outperforming it significantly in `PenGoal-v1` and `Claw-v1`. In principle,

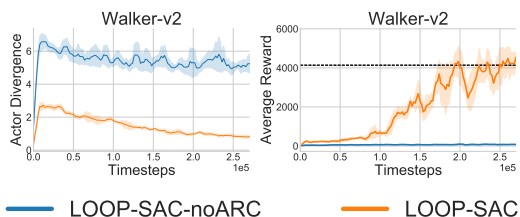

Figure 4: (Left) ARC reduces the actor-divergence measured by the L2 distance between the mean of the parametrized actor and the output of the H-step lookahead policy. (Right) In absence of ARC, policy learning can be unstable.

methods like MBPO and value expansion can be combined with LOOP to potentially increase performance; we leave such combinations for future work. LOOP-SARSA has poor performance as a result of the poor value function that is trained for evaluating replay buffer policy rather than optimality. As an ablation study, we also run experiments using LOOP without ARC, which optimizes the unconstrained objective of Eqn. 3 using CEM [36]. Figure 4 (left) shows that ARC reduces actor-divergence effectively and Figure 4 (right) shows that learning performance is poor in absence of ARC for `Walker-v2`. More ablation results can be found in Appendix D.5.

## 6.2 LOOP for Offline RL

| Dataset | Env | CRR | LOOP CRR | Improve% | PLAS | LOOP PLAS | Improve% | MBOP |
|---------|-----|-----|----------|----------|------|-----------|----------|------|
| medium | hopper | 65.73 | **85.83** | 30.6 | 32.08 | 56.47 | 76.0 | 48.8 |
| | halfcheetah | 41.14 | **41.54** | 1.0 | 39.33 | 39.54 | 0.5 | **44.6** |
| | walker2d | 69.98 | **79.18** | 13.1 | 46.20 | 52.66 | 14.0 | 41.0 |
| med-replay | hopper | 27.69 | 29.08 | 5.0 | 29.29 | **31.29** | 6.8 | 12.4 |
| | halfcheetah | 42.29 | 42.84 | 1.3 | 43.96 | **44.25** | 0.7 | 42.3 |
| | walker2d | 19.84 | 27.30 | 37.6 | 35.59 | **41.16** | 15.7 | 9.7 |

Table 1: Normalized scores for LOOP on the D4RL datasets comparing to the underlying offline RL algorithms and a baseline MBOP. LOOP improves the base algorithm across various types of datasets and environments.

For Offline RL, we benchmark the performance using the D4RL datasets [53]. We combine LOOP with two value-based offline RL algorithms: Critic Regularized Regression (CRR) [54] and Policy in Latent Action Space (PLAS) [32]. We use the original offline RL algorithms to train a value function from the static data and then use it as the terminal value function for LOOP. We use $\beta = 1$ in the trajectory prior of ARC (Section 5.1) in the offline RL setting to keep the policy conservative.

**Baselines:** In addition to the underlying offline RL algorithms, we also include recent work MBOP [33] as a baseline. MBOP uses a terminal value function which is an evaluation of the dataset policy. In contrast, LOOP uses a terminal value function trained with offline RL algorithms which is more optimal.

**Performance:** Table 1 presents the comparison of LOOP and the underlying offline RL algorithms. LOOP offers an average improvement of 15.91% over CRR and 29.49% over PLAS on the complete D4RL MuJoCo Locomotion dataset. Full results can be found in Appendix D.3. The results further highlight the benefit of the LOOP framework compared to the underlying model-free algorithms.

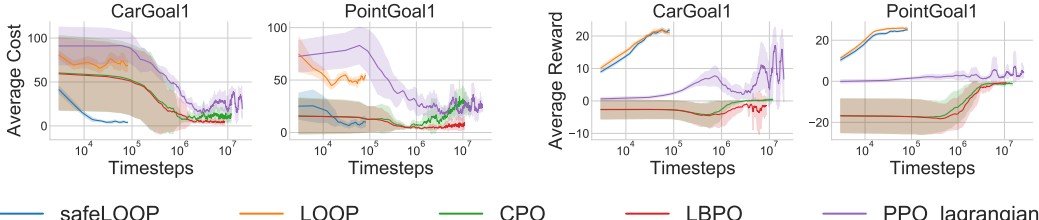

Figure 5: We compare safeLOOP with other safety methods such as CPO, LBPO, and PPO-lagrangian on OpenAI Safety Gym environments. It shows significant sample efficiency while offering similar or better safety benefits as the baselines.

## 6.3 LOOP for Safe RL

For safe RL, we modify the H-step lookahead optimization to maximize the sum of rewards while satisfying the cost constraints, as described in Section 5.2. We evaluate our method on two environments from the OpenAI Safety Gym [55] and an RC-car simulation environment [56]. The objective of the Safety Gym environments is to move a Point mass agent or a Car agent to the goal while avoiding obstacles. The RC-car environment is rewarded for driving along a circle of 1m fixed radius with a desired velocity while staying within the 1.2m circle during training. Details for the environments can be found in Appendix C.4.

**Baselines**: We compare our safety-augmented LOOP (safeLOOP) against various state-of-the-art safe learning methods such as CPO [57], LBPO [58], and PPO-lagrangian [59, 55]. CPO uses a trust region update rule that guarantees safety. LBPO relies on a barrier function formulated around a Lyapunov constraint for safety. PPO-lagrangian uses dual gradient descent to solve the constrained optimization. To ensure a fair comparison, all policies and dynamics models are randomly initialized, as is commonly done in safe RL experiments (rather than starting from a safe initial policy). We additionally compare against a model-based safety method that modifies PETS for safe exploration (safePETS)

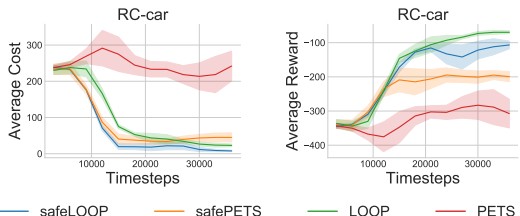

Figure 6: RC-car experiments show the importance of the terminal value function in the LOOP framework. SafeLOOP achieves higher returns than safePETS while being competitive in safety performance. Both safePETS and PETS fail to learn a drifting policy due to limited lookahead.

without the terminal value function. We mostly compare to model-free baselines due to a lack of safe model-based Deep-RL baselines in the literature.

**Performance:** For the OpenAI Safety Gym environments, we observe in Figure 5 that safeLOOP can achieve performant yet safe policies in a sample efficient manner. SafeLOOP reaches a higher reward than CPO, LBPO and PPO-lagrangian, while being orders of magnitude faster. SafeLOOP also achieves a policy with a lower cost faster than the baselines. From another aspect, the simulated RC-car experiments demonstrate the benefits of the terminal value function in safe RL. Figure 6 shows the performance of LOOP, safeLOOP, PETS, and safePETS on this domain. PETS [17] and safePETS do not consider a terminal value function. SafeLOOP is able to achieve high performance while maintaining the fewest constraint violations during training. Qualitatively, LOOP and safeLOOP are able to learn a safe drifting behavior, whereas PETS and safePETS fail to do so since drifting requires longer horizon reasoning beyond the fixed planning horizon in PETS. The results suggest that safeLOOP is a desirable choice of algorithm for safe RL due to its sample efficiency and the flexibility of incorporating constraints during deployment.

## 7 Conclusion

In this work we analyze the H-step lookahead method under a learned model and value function and demonstrate empirically that it can lead to many benefits in deep reinforcement learning. We propose a framework LOOP which removes the computational overhead of trajectory optimization for value function update. We identify the actor-divergence issue in this framework and propose a modified trajectory optimization procedure - Actor Regularized Control. We show that the flexibility of H-step lookahead policy allows us to improve performance in online RL, offline RL as well as safe RL and this makes LOOP a strong choice of RL algorithm for robotic applications.

**Acknowledgments**

We thank Tejus Gupta, Xingyu Lin and the members of R-PAD lab for insightful discussions. This material is based upon work supported by the United States Air Force and DARPA under Contract No. FA8750-18-C-0092, LG Electronics, and the National Science Foundation under Grant No. IIS-1849154.

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
