# OpenReview forum: "Learning Off-Policy with Online Planning"
_robot-learning.org/CoRL/2021/Conference — CoRL2021 Oral_

### Official Review · Reviewer_ztVy · 2021-07-22

**Originality:** Good
**Technical Quality:** Very Good
**Clarity Of Presentation:** Very Good
**Impact:** 4

**Recommendation:**

Strong Accept: I recommend accepting the paper and will argue for my recommendation even if other reviewers hold a different opinion.

**Summary:**

The paper presents an RL approach that combines model-based and model-free learning and improves upon previous H-step lookahead methods. In particular, H-step lookahead methods employ a model to perform trajectory optimization for H steps and then use a learned value function to estimate terminate values of the trajectories, which enables infinite-horizon optimization. The paper first performs a theoretical analysis of the induced errors of model-based and value learning parts and concludes that there is an advantage of combining them, especially in low data regimes. After that, the paper identifies the problem of “actor divergence”, which is present in previous methods. In particular, as the behavior policy used to collect the data differs from the parameterized actor used to optimize the value function, it can lead to bootstrapping errors. The paper proposes to introduce a KL-divergence constraint on the behavior policy optimization that constraints it against the actor policy and in this way reduces the actor divergence. Finally, the paper introduces ensemble-based uncertainty estimation for incorporating pessimism when learning from offline data, and points out the ability to incorporate safety constraints when performing model-based trajectory optimization. The method is evaluated in multiple simulated MuJoCo tasks, such as OpenAI Gym tasks for Online RL evaluations, D4RL for Offline RL, and OpenAI Safe Gym for Safe RL evaluations. It is shown to outperform multiple baselines and reduce the actor divergence problem.

**Issues:**

Please refer to "Strengths And Weaknesses" for a detailed list of issues.

**Reviewer Expertise:**

Good: General knowledge of the area

**Strengths And Weaknesses:**

Strengths:
- H-step lookahead methods are an elegant way to combine advantages of model-based and model-free learning. The paper presents an important improvement for this family of methods.
- Theoretical error analysis is a good way to demonstrate advantages of H-step lookahead methods.
- The paper is well-written and easy to understand and follow.
- The method is shown to outperform a range of baselines in both online and offline problems in simulation, as well as work with simulated safety-critical tasks.

Weaknesses:
- Although mentioning POLO in the related work, experiments do not include this method as a baseline. It would be useful to see comparisons to this method as it is close to the presented approach and would help to evaluate the advantage of reducing the actor divergence.
- It would be interesting to see an ablation of adding the pessimistic term to optimization to better understand its impact.
- It is not completely clear how a suboptimal actor would affect the behavior policy in case the actor optimization does not work well. It would be interesting to see a discussion on this aspect of the algorithm.

-- Post rebuttal --
Thanks for clarifications and additional experiments, I have increased my score.


**Summary Of Recommendation:**

The paper improves upon previous works on H-step lookahead methods and provides a range of experiments and theoretical derivations. Some improvements could be made through additional experiments and discussions.

---

> ### Author Response · Authors · 2021-08-29
> **Response to Reviewer ztVy**
>
> Thank you for the detailed review of our paper and encouraging comments about the approach and results. We address your questions below. Please let us know if further clarification is needed.
>
> > Although mentioning POLO in the related work, experiments do not include this method as a baseline. It would be useful to see comparisons to this method as it is close to the presented approach and would help to evaluate the advantage of reducing the actor divergence.
>
> According to the suggestion, we add the curves of POLO for Claw-v1 and HalfCheetah-v2 in Figure 9 of Appendix D.2. The author's implementation of POLO is unavailable so we tried our best to implement it. To have a fair comparison with LOOP, we keep the hyperparameters as close to LOOP as possible and use a learned model; since the code for POLO is not available, we are unsure what hyperparameters were used in the original experiments. The performance of POLO is pretty low compared to LOOP, potentially due to the limited computation of MPPI used for the value function update. Potentially the performance of POLO would be better with much larger computational resources than what we have available.
>
> We would also like to highlight the difference in computational efficiency in LOOP and POLO. POLO requires an additional trajectory optimization procedure for value function computation, which is very computationally expensive. In contrast, LOOP learns a parameterized policy and value function to make the value function computation significantly faster.  In addition, normally for online planning, we can warmstart from the results from the previous time step ("amortization"); LOOP and PETS [3] take advantage of this optimization.  In contrast, POLO cannot take advantage of this amortization during optimization for the value computation because it samples states IID from the replay buffer. Our implementation of POLO (after reasonable optimizations) takes ~84 hours for 100k steps of HalfCheetah on a single NVIDIA 1080 GPU whereas LOOP takes ~7 hours (12x less computation) while taking ~1/5 the memory consumption of POLO.
>
> > It would be interesting to see an ablation of adding the pessimistic term to optimization to better understand its impact.
>
> First, we would like to clarify that the pessimistic term is not itself one of our contributions; this pessimistic term was used in previous works in model-based offline RL like [1], [2] (references below) which learn a policy given the data in an uncertainty-penalized MDP.  Nonetheless, we add an ablation in Table 6, Appendix D.4 for the pessimistic term in the Offline RL experiments. We observe that pessimistic term is useful for Offline RL to prevent incorrect extrapolation.
>
> > It is not completely clear how a suboptimal actor would affect the behavior policy in case the actor optimization does not work well. It would be interesting to see a discussion on this aspect of the algorithm.
>
> That is a great point. Theorem 1 implicitly subsumes the error resulting from a suboptimal parameterized actor as value errors. Our analysis shows that the H-step lookahead policy is more robust to value errors compared to the original suboptimal actor.
>
> References:
>
> [1]: [MOPO: Model-based Offline Policy Optimization](https://arxiv.org/abs/2005.13239)
>
> [2]: [MOReL : Model-Based Offline Reinforcement Learning](https://arxiv.org/abs/2005.05951)
>
> [3]: [Deep Reinforcement Learning in a Handful of Trials using Probabilistic Dynamics Models](https://arxiv.org/abs/1805.12114)

---

> > ### Comment · Reviewer_ztVy · 2021-08-30
> > **Thank you!**
> >
> > Thanks for clarifications and additional experiments, I have increased my score.

---

### Official Review · Reviewer_Sv1h · 2021-07-23

**Originality:** Good
**Technical Quality:** Excellent
**Clarity Of Presentation:** Very Good
**Impact:** 3

**Recommendation:**

Strong Accept: I recommend accepting the paper and will argue for my recommendation even if other reviewers hold a different opinion.

**Summary:**

This paper proposes a novel approach to reinforcement learning with planning.
The idea is to learn a model of the dynamics and

1. Use a planning algorithm to find the optimal sequence of actions in the H-horizon.
2. Use at the last step of the horizon an approximation of $V^*$, so to make the planning algorithm aware of the value beyond the horizon.
3. The value function is learned with an off-policy reinforcement learning algorithm
4. Since the policy of the trajectory planner differs from the policy induced by the off-policy RL, the authors propose to integrate in the planner a penalization term for actions that are far from the policy of the reinforcement learning counterpart.

In particular, the authors propose bounds on the return of the policy induced by the planner by both taking into consideration the error from the model and from the value function.

The authors propose both a version of their algorithm for offline RL and for constrained MDP.



**Issues:**

- I would like to see Theorem 1 with also an assumption on suboptimal trajectory planners.
- I would like a discussion on the difference between providing $V^*$ instead of $V^\pi$ to the planner.

**Reviewer Expertise:**

Very good: Comprehensive knowledge of the area

**Strengths And Weaknesses:**

__Strenghts__

The idea of integrating the value function into the planner to extend the horizon is not new [1, 2]. The merits of the paper are, in my opinion,

- the theoretical analysis
- the introduction of the ARC (the Actor Regularized Control).
- the usage of the algorithm in both offline RL and constrained MDP.

I think that the paper is well written and follows a good argumentation of the proposed method.

When I read the paper, I immediately thought that the RL algorithm should only evaluate the policy, rather than improve it, (e.g., should evaluate $V^\pi$ instead of $V^*$). Doing that should avoid the problem that you mention of having a policy in the RL algorithm that differs from the policy of the planner. I saw in the experiment that you actually did that with LOOP-SARSA. I see that LOOP-SARSA does not obtain a good performance (I think that this is due to the fact that it does not "bootstrap" the optimal value function). It would be nice still to insert a more detailed explanation of why you choose to estimate $V^*$ instead of $V^\pi$ (theoretically, the improvement is still guaranteed by the fact that the planner seeks optimal actions - this is the same principle driving behind SARSA).

The idea of keeping the trajectory close to the one generated by the RL agent does make sense too, otherwise, there is a distribution mismatch.

I think that the empirical section is well made, there are enough comparisons with related algorithms.


__Weakness__

The bound (eq 5) assumes that the planner is picking optimal actions. In reality, this is not the case. Therefore, the authors should insert a term that describes the situation where the planner plans a suboptimal trajectory.

I would insert the discussion on whether to use policy evaluation (estimate $V^\pi$) or policy improvement (estimate $V^*$). What are the pros and cons of the two approaches?



__typo__

in line 195 the inline math has an extra parenthesis.

Update
==

I appreciated the authors including my suggestion in the paper. After reading the other reviews, I am willing to keep my score unchanged.


__References__

[1] https://ieeexplore.ieee.org/stamp/stamp.jsp?tp=&arnumber=6614995
[2] https://arxiv.org/pdf/2102.11122.pdf

**Summary Of Recommendation:**

I do not think that the idea presented in the paper is very novel (planning + RL is not novel, (value-function to extend the horizon is not novel), (regularizing the actor is also not a novel concept)...
Still, the paper is well written, and all the choices are, in my opinion, fair and logical. Also, each contribution of the paper might be small but summed all together it makes a fair level of originality.

The authors did a good job in comparing their algorithm with related work.
Therefore, I advocate for the paper's acceptance.

---

> ### Author Response · Authors · 2021-08-29
> **Response to Reviewer Sv1h**
>
> Thank you for the detailed review and summarization of our paper and encouraging comments about the approach and results. The review brings up some interesting points which we discuss below. Please let us know if further clarification is needed.
> > When I read the paper, I immediately thought that the RL algorithm should only evaluate the policy, rather than improve it, (e.g., should evaluate v_pi instead of v*). Doing that should avoid the problem that you mention of having a policy in the RL algorithm that differs from the policy of the planner. I saw in the experiment that you actually did that with LOOP-SARSA. I see that LOOP-SARSA does not obtain a good performance (I think that this is due to the fact that it does not "bootstrap" the optimal value function). It would be nice still to insert a more detailed explanation of why you choose to estimate v* instead of v_pi (theoretically, the improvement is still guaranteed by the fact that the planner seeks optimal actions - this is the same principle driving behind SARSA).
>
> To clarify, LOOP-SARSA is evaluating the "replay buffer policy" instead of the H-step lookahead policy because we are using off-policy data (where the original SARSA is an on-policy algorithm). This is probably why it doesn't work well. Unfortunately, on-policy LOOP-SARSA would be too slow, due to the need for collecting on-policy data.  POLO is actually exactly evaluating $V^\pi$. However, POLO requires running trajectory optimization during the value function update, which is computationally expensive. In contrast to these methods, LOOP uses an off-policy algorithm to learn  $V^*$, as the reviewer noted. We found that this approach has good performance and it is significantly more computationally efficient than POLO. An interesting direction of future work could be to try to combine LOOP with an efficient off-policy evaluation algorithm to estimate $V^\pi$.
>
> > The bound (eq 5) assumes that the planner is picking optimal actions. In reality, this is not the case. Therefore, the authors should insert a term that describes the situation where the planner plans a suboptimal trajectory.
>
> Thank you for the suggestion! We have modified Theorem 1 to reflect this suboptimality. We observe that H-step lookahead scales linearly with the H-step optimization suboptimality.
>
> > in line 195 the inline math has an extra parenthesis.
>
> Thank you for pointing this out. We have fixed the typo in the new version of the paper.

---

> > ### Comment · Reviewer_Sv1h · 2021-08-30
> > **Thank you for your response**
> >
> > Dear authors,
> >
> > thanks for considering my suggestion in Theorem 1, and for your rebuttal.
> > I have read also the other reviews, and I will keep my score as it is.
> >
> > Best regards.

---

### Official Review · Reviewer_PXsM · 2021-07-24

**Originality:** Very Good
**Technical Quality:** Excellent
**Clarity Of Presentation:** Very Good
**Impact:** 4

**Recommendation:**

Strong Accept: I recommend accepting the paper and will argue for my recommendation even if other reviewers hold a different opinion.

**Summary:**

This paper presents a theoretical analysis and practical implementation of a learning framework that combines H-step lookahead policies with a learned model and a learned terminal value function using a model-free off-policy algorithm. This method, called Learning Off-Policy with Online Planning (LOOP), reduces the computational cost of the trajectory optimization by updating the value function through using a model-free off-policy algorithm. In addition, the authors propose a modified trajectory optimization method called Actor Regularized Control to address the Actor Divergence issue (different policies used for data collection and to learn the value function). Moreover, the authors show the LOOP framework can be applied to enhance performance in offline RL and safe RL tasks. Finally, the method is tested on different simulated tasks, where it is shown to provide performance improvements for online RL, offline reinforcement learning and safe RL when compared with state-of-the-art methods.

**Issues:**

There are no issues to be addressed for this paper.

**Reviewer Expertise:**

Good: General knowledge of the area

**Strengths And Weaknesses:**

Strengths
- The paper is well written and clear to understand. The authors do a good job introducing the state of the art methods related with their work and how their work differentiates from others. The authors also present in a clear and structured way each of the different contributions of the paper and they provide a detailed analysis of the extensive testing of their framework and comparison with other methods.
- A theoretical analysis is presented to provide a solid foundation for the method and back up the experimental results.
- It is interesting to see how this framework can be adapted not only for online RL, but also for offline RL and safe RL, which makes it versatile and appealing for its application in robotic tasks.

Weaknesses
- Despite the fact that a model-free off-policy algorithm is used to compute the Value function, including the H-step lookahead policy may still increase the computational cost of the algorithm as the action is selected though trajectory optimization by rolling out the dynamic model to find an action sequence with the highest return.


**Summary Of Recommendation:**

I would recommend the authors to include a section to discuss the application of its framework to high-dimensional complex robotics tasks e.g. humanoid and what could be the potential issues or advantages. Though they have shown their method performs as good or better than baselines available for comparison, it would be interesting to see its application in 3d complex environments and potential real robotics applications.

---

> ### Author Response · Authors · 2021-08-29
> **Response to Reviewer PXsM**
>
> Thank you for the detailed review of our paper and encouraging comments about the approach and results. We address your two questions below. Please let us know if further clarification is needed.
>
> > Despite the fact that a model-free off-policy algorithm is used to compute the Value function, including the H-step lookahead policy may still increase the computational cost of the algorithm as the action is selected through trajectory optimization by rolling out the dynamic model to find an action sequence with the highest return.
>
> The underlying off-policy algorithm (SAC) can run policy inference at ~1400 Hz in our experiments. LOOP does have a higher computation cost due to the additional optimization in the H-step lookahead and is able to run at ~20Hz on our 1 GPU machine, which is a reasonable speed for many real robot applications. We speed up this additional trajectory optimization by warm-starting with the solutions from the previous timestep (usually referred to as "amortization"). We would also like to highlight that the H-step lookahead can afford us other benefits like safety and interpretability, which perhaps makes it worth the additional computation.
>
> > I would recommend the authors to include a section to discuss the application of its framework to high-dimensional complex robotics tasks e.g. humanoid and what could be the potential issues or advantages. Though they have shown their method performs as good or better than baselines available for comparison, it would be interesting to see its application in 3d complex environments and potential real robotics applications.
>
> Following the suggestion, we ran the method on the Humanoid environment to further test our method in an environment with a high-dimensional state and action space, and we add the additional results in the paper to Appendix D.1 and Figure 8.  As shown, LOOP-SAC shows a significant improvement over SAC, which is consistent with our results on other environments. We will try to run the other baselines for this environment for the final version of the paper. Note that we are using a TruncatedHumanoid environment following previous papers [1][2][3] (references listed below).
>
> On the topic of applying LOOP to high-dimensional problems, we would like to highlight that LOOP works well on the PenGoal environment which has an observation space of 45-dim and action space of 24-dim (similar to TruncatedHumanoid which has an observation space of 45-dim and action space of 17-dim). As shown in Figure 3, LOOP-SAC significantly outperforms the other methods on the PenGoal environment.
> In general, with a higher dimensional observation space, LOOP might need a higher capacity dynamics model; with a high-dimensional action space, there could be increasing computational cost in trajectory optimization. Nonetheless, our experiments still show LOOP-SAC outperforming SAC in both of these environments (TruncatedHumanoid and PenGoal).
>
> References:
>
>  [1] : [When to Trust Your Model: Model-Based Policy Optimization](https://arxiv.org/abs/1906.08253)
>
>  [2]: [Trust the Model When It Is Confident: Masked Model-based Actor-Critic](https://arxiv.org/abs/2010.04893)
>
>  [3]: [Model-based Policy Optimization with Unsupervised Model Adaptation](https://arxiv.org/abs/2010.09546)

---

### Meta-Review · Area_Chair_7JZQ · 2021-08-03

**Recommendation:** Accept (Oral)
**Confidence:** 4

**Metareview:**

=== comments before the discussion ===

The reviewers agree that the paper is well-written and the technical quality is good. On the other hand, reviewers provided comments to improve the paper, e.g., adding ablation study and missing discussions. I recommend the authors to carefully address the comments provided by reviewers.

=== comments after the discussion ===

The paper present a novel RL approach that combines model-based and model-free learning. As the scores provided by reviewers are very high, the area chair recommends the acceptance of the paper (oral).

---

> ### Author Response · Authors · 2021-08-29
> **Response to Area Chair 7JZQ**
>
> We thank all the reviewers and meta-reviewer for reviews and interesting points that were brought up in the reviews. We address the questions in individual replies. Please let us know if further clarifications are needed. Here is a summary of all the changes to the paper and supplementary material:
>
> * **Assumption on an optimal planner**: We originally assumed an optimal planner for H-step lookahead for LOOP, but as reviewer Sv1h pointed out, the planner is not generally optimal in practice. We update our proof for Theorem 1 to take this suboptimality into account.
>
> * **Comparison with POLO**: We add a comparison with POLO on one locomotion and one manipulation task (Supplementary Figure 9) while highlighting the computational difficulties encountered in running the experiments in Appendix D.2. For comparison, we modify POLO to use a learned dynamics model similar to LOOP (instead of using the ground-truth dynamics model as in the original POLO paper) and we observe poor performance for POLO after training to the best of our abilities.
>
> * **Choice of the terminal value function**: We add to Appendix B.4 a discussion about choosing V^* vs V^pi as the terminal value function.
>
> * **Ablation of pessimism parameter in Offline-RL experiments**: We add ablation results of the pessimism parameter in Appendix D.4.
>
> * **Fixed typos at Line 195**. Changes to the paper based on suggestions have been highlighted in red in the new version of the paper.

---

### Decision · Program_Chairs · 2021-09-13

**Decision:**

Accept (Oral)

**Comment:**

=== comments before the discussion ===

The reviewers agree that the paper is well-written and the technical quality is good. On the other hand, reviewers provided comments to improve the paper, e.g., adding ablation study and missing discussions. I recommend the authors to carefully address the comments provided by reviewers.

=== comments after the discussion ===

The paper present a novel RL approach that combines model-based and model-free learning. As the scores provided by reviewers are very high, the area chair recommends the acceptance of the paper (oral).